# Metabolic Signatures: Pioneering the Frontier of Rectal Cancer Diagnosis and Response to Neoadjuvant Treatment with Biomarkers—A Systematic Review

**DOI:** 10.3390/ijms25042381

**Published:** 2024-02-17

**Authors:** Răzvan Alexandru Ciocan, Andra Ciocan, Florin Vasile Mihăileanu, Cristina-Paula Ursu, Ștefan Ursu, Cătălin Bodea, Ariana-Anamaria Cordoș, Bogdan Augustin Chiș, Nadim Al Hajjar, Noemi Dîrzu, Dan-Sebastian Dîrzu

**Affiliations:** 1Department of Surgery-Practical Abilities, “Iuliu Hațieganu” University of Medicine and Pharmacy, 400337 Cluj-Napoca, Romania; Razvan.Ciocan@umfcluj.ro; 2Department of Surgery, “Iuliu Hațieganu” University of Medicine and Pharmacy, 400162 Cluj-Napoca, Romania; mihaileanu@umfcluj.ro (F.V.M.); cristinapaulapop10@yahoo.com (C.-P.U.); stefan.ursu20@gmail.com (Ș.U.); bodea_cata@yahoo.com (C.B.); nadim.alhajjar@umfcluj.com (N.A.H.); 3“Prof. Dr. Octavian Fodor” Regional Institute of Gastroenterology and Hepatology, 400162 Cluj-Napoca, Romania; 4Romanian Society of Medical Informatics, 300041 Timisoara, Romania; cordos.ariana@umfcluj.ro; 5Department of Internal Medicine, “Iuliu Hațieganu” University of Medicine and Pharmacy, 400162 Cluj-Napoca, Romania; chis.augustin@umfcluj.ro; 6Clinical Laboratory Department, Transilvania Hospital, 400486 Cluj-Napoca, Romania; 7Emergency County Hospital Cluj, 400006 Cluj-Napoca, Romania; dirzudan@gmail.com; 8STAR—UBB Institute, Babeș Bolyai University, 400084 Cluj-Napoca, Romania

**Keywords:** rectal cancer, metabolomics, biomarkers, neoadjuvant treatment

## Abstract

Colorectal cancer (CRC) is one of the most aggressive, heterogenous, and fatal types of human cancer for which screening, and more effective therapeutic drugs are urgently needed. Early-stage detection and treatment greatly improve the 5-year survival rate. In the era of targeted therapies for all types of cancer, a complete metabolomic profile is mandatory before neoadjuvant therapy to assign the correct drugs and check the response to the treatment given. The aim of this study is to discover specific metabolic biomarkers or a sequence of metabolomic indicators that possess precise diagnostic capabilities in predicting the efficacy of neoadjuvant therapy. After searching the keywords, a total of 108 articles were identified during a timeframe of 10 years (2013–2023). Within this set, one article was excluded due to the use of non-English language. Six scientific papers were qualified for this investigation after eliminating all duplicates, publications not referring to the subject matter, open access restriction papers, and those not applicable to humans. Biomolecular analysis found a correlation between metabolomic analysis of colorectal cancer samples and poor progression-free survival rates. Biomarkers are instrumental in predicting a patient’s response to specific treatments, guiding the selection of targeted therapies, and indicating resistance to certain drugs.

## 1. Introduction

Colorectal cancer (CRC) is the third most often detected malignancy and the fourth highest contributor to cancer-related deaths in both males and females globally [1]. Rectal cancer (RC) comprises approximately 30% of colorectal cancer cases and is linked to a poor clinical prognosis. Neoadjuvant chemoradiation (nCRT) is the established approach for managing locally advanced rectal cancer. This treatment is followed by a whole mesorectal excision to enhance the capacity to surgically remove the tumor, preserve the anal sphincter, and achieve effective local control [2,3].

Early-stage (in situ carcinoma) detection and treatment greatly improve the 5-year survival rate, potentially reaching 90% [4]. Furthermore, there are distinct variations in the distribution of lesions between RC and colon cancer (CC). Once the CRC tumor identity is verified by both endoscopic and histopathologic investigations, distinct approaches should be utilized for the treatment of RC compared to CC. Hence, it is crucial to differentiate between these two categories of tumors [5].

The National Comprehensive Cancer Network (NCCN) recommendation for rectal cancer screening identifies fecal occult blood tests and colonoscopy as the primary screening methods [5,6]. Nevertheless, these two approaches have limitations in terms of low adherence, incomplete diagnosis, and inaccurate positive results [7]. Currently, tumor markers such as carcinoembryonic antigen (CEA) and carbohydrate antigen 19-9 (CA 19-9) are often employed for non-invasive screening. Nevertheless, these two serum indicators have very low sensitivity and specificity, resulting in elevated rates of false negatives. They are less effective in diagnosing asymptomatic individuals. Furthermore, these two markers are incapable of distinguishing between CC and RC. Therefore, it is imperative to discover effective, non-intrusive, and precise indicators for rectal cancer screening [8].

Metabolomics is a developing area of study that follows transcriptomics, genomes, and proteomics. It primarily focuses on the comprehensive analysis of various components in biological fluids, tissues, and cell extracts. Currently, it serves as a research model in several domains, such as diagnosis [9,10], biomarker screening [11,12], nutritional intervention [13], and chemical safety evaluation [14,15]. Three widely used analytical methods for assaying and quantifying metabolites are liquid chromatography (LC) combined with mass spectrometry (MS), gas chromatography MS (GC/MS), and nuclear magnetic resonance (NMR) [16]. Thus far, only a few investigations utilizing biological rectal cancer samples and NMR techniques have been documented [17,18]. Nevertheless, the research conducted had a restricted number of patient’s tissues and hence failed to yield precise and all-encompassing data on RC metabolites. Furthermore, there has been little evidence of the identification of specific metabolites that are associated with the various stages of rectal cancer. Hence, doing metabolic profiling of human rectal cancer tissues will be beneficial in facilitating molecular diagnostics and offering fresh perspectives on rectal cancer.

The aim of this study is to discover specific metabolic biomarkers or a sequence of metabolomic indicators which possess precise diagnostic capabilities in predicting the efficacy of neoadjuvant therapy.

Research questions:Which are the metabolic molecules with potential predictive value for neoadjuvant treatment responses?What role does every metabolite play in understanding the molecular changes occurring in rectal tumor tissue prior and after the oncological treatment?

## 2. Materials and Methods

The approach utilized in this systematic review consisted of developing search algorithms, criteria for selection, and protocols for data extraction. Adherence to the Preferred Reporting Items for Systematic Reviews and Meta-Analyses (PRISMA) statement requirements was maintained, as demonstrated in Figure 1. The search encompassed a timespan from January 2013 to December 2023, specifically targeting articles published in English language and available through online databases like PubMed (Medline), Embase, and Clarivate Web of Science.

The search approach employed the keywords “rectal cancer”, “metabolomics”, and “biomarkers” using the Boolean Operator “AND” to connect each of these terms. During the designated period from 2013 to 2023, papers with English titles were evaluated for eligibility based on their title and abstract. Two researchers conducted the assessment to remove any duplicate articles. In addition, a manual search was performed by examining the reference lists of the identified articles and reviews to discover any possible studies that were not included in the original search.

Inclusion criteria: All papers that provided information on the examination of metabolic markers in rectal cancer tumors were deemed suitable for inclusion. The eligible categories of research were original articles, clinical trials, and randomized control trials. All the above-mentioned types of articles had to be open access. The studies selected involved human subjects.

Exclusion criteria: The present study excluded articles published before 2013, lacked relevance to the specified subject, letters to the editor, short reports, meta-analyses, systematic reviews, narrative reviews, non-English papers, studies involving patients under 18 years old, and articles solely focused on proteomics, genomics, or rectal cancer as an isolated entity. Moreover, articles involving non-human subjects or in vitro studies were excluded. Additionally, studies addressing the application of metabolomics in benign rectal disorders and acute inflammatory processes were also excluded.

Following the exploration of the specified keyword combination, a total of 108 articles providing data on the identification of metabolic biomarkers for rectal cancer were identified. Within this set, one article was excluded due to being written in a non-English language, and seven were disregarded as they fell outside of the 10-year timeframe. Removal of duplicated articles, publications not pertinent to the subject, systematic reviews, and papers with open access restrictions resulted in seven scientifically relevant papers qualifying for inclusion in this study (Figure 1).

For a comprehensive assessment of bias risk in each study, we employed the Quality Assessment of Diagnostic Accuracy Studies-2 (QUADAS-2) revised tool (Accessed on 1 December 2023), comprising four bias domains: patient selection, index test, reference standard, and patient flow and timing [19]. A Microsoft Excel, Microsoft Office 2021 extraction tool was utilized in order to organize and collect data. Two independent reviewers conducted evaluations on all selected studies, with consensus-based final judgments involving a third reviewer, when necessary, in order to thoroughly select the appropriate papers to be included in this research.

## 3. Results

The seven studies included in the review encompass relevant biomarkers that allow early detection of RC formation, invasion, and metastasis [20,21,22,23,24,25]. In 2013, Wang et al. [22] applied 1H-NMR to study metabolic profiling of human rectal cancer tissues. The researcher found metabolic alterations between rectal cancer tissues and normal controls. A total of 38 differential metabolites were identified, 16 of which were closely correlated with the stages of rectal cancer.

In 2016, Redalen et al. [23] was approved by the Institutional Review Board, The Regional Committee for Medical and Health Research Ethics of Southeast Norway, and The Norwegian Medicines Agency. It included 54 patients enrolled between October 2005 and May 2008, in comparison to Jia et al. [21], who enrolled 106 patients in their research in 2018. They were diagnosed with clinical T3-4 and/or N+ rectal cancer without distant metastasis at Fudan University Shanghai Cancer Center between July 2014 and January 2016. Wu et al. [24] obtained approval from the Ethics Committee of the First Affiliated Hospital of Zhejiang University. They admitted patients to the hospital between December 2016 and May 2017, which included 22 CC patients and 23 RC patients. In 2022, Strybel et al. [25] published their research. There were 40 patients included in that study, all diagnosed with rectal adenocarcinoma. They underwent neoadjuvant radiation in a total dose of 39 to 54 Gy, completed in 8 to 104 days before surgery.

The average total number of RC patients investigated was 75.5 ± 14.40 (38 [47.25–95.75]) with a dominance across all studies of the male patients. Two of the articles provide information about the tumor distances from the anal verge [20,21]. The available patient demographics are represented in Table 1.

Regarding the samples used and method applied in the second study by Redalen et al. [23], at the time of diagnosis, five tumor biopsies were collected, snap-frozen in liquid nitrogen, and stored at −80 °C. The high-resolution magic angle spinning magnetic resonance spectroscopy (HR MAS MR) spectra were acquired on a Bruker Avance DRX600 spectrometer (Bruker, Billerica, MA, USA) equipped with a 1H/13C MAS probe with a gradient aligned with the magic angle axis. In the third study by Jia et al. [21], at the time of diagnosis, 3–5 tumor biopsies were collected, snap-frozen in liquid nitrogen, and stored at −80 °C. Tumor metabolic profiles were generated by HR MAS MRS. In the fourth study, by Wu et al., fasting blood samples were collected and serum was isolated for further analysis. The researchers used an Agilent 7890A GC coupled to an Agilent 5975B MSD (Agilent, Santa Clara, CA, USA) with an electron ionization (EI) source for chromatographic separation. They performed peak detection with R version 3.3.2 software [24]. Crotti et al. [20] involved rectal cancer patients, who were due to receive preoperative chemoradiation. Healthy rectal mucosa and pre-therapy tumor biopsies were processed to isolate total RNA by TRIzol™ reagent (Invitrogen, Waltham, MA, USA). Mass spectrometry measurements were performed using an API 4000 triple quadrupole mass spectrometer. Small extracellular vesicles (EVs) were isolated in the fifth study from the serum of patients with rectal cancer by the size exclusion chromatography (SEC) method. Known exosomal proteins, CD9, CD63, CD81, ALIX, and TSG101 were analyzed in EV fractions by Western blot technique. The concentration of isolated PKH67-labeled EVs fraction was measured by Amnis ImageStream^®^X Mark II (Merck, Darmstadt, Germany).

Table 2 provides the exhaustive list of biomarkers proving predictive values in the selected studies.

In the research published in Mol. Cancer [22], a panel of metabolites was identified to be significantly different between rectal cancer tissues and normal controls. These metabolites were involved in the metabolic pathways of glycolysis, serine synthesis, the tricarboxylic acid cycle, amino acid metabolism, pyrimidine metabolism, and gut flora metabolism. Lactate levels were significantly increased in RC tissues compared to normal controls. This is consistent with the Warburg effect, which is a metabolic phenomenon in cancer cells that leads to the preferential use of glycolysis for energy production. Myo-inositol levels were significantly decreased in RC tissues compared to normal controls. The decreased levels of myo-inositol may contribute to the development of cancer-related symptoms. Glutathione levels were significantly decreased in RC tissues compared to normal controls. The decreased levels of glutathione may make cells more susceptible to oxidative stress and tumor progression. Formate levels were significantly increased in RC tissues compared to normal controls. The increased levels may contribute to the development of cancer-related symptoms. In contrast, the article published in Rectal Cancer [23] suggests that high tumor concentrations of glycine, creatine, and myo-inositol may be useful biomarkers for predicting poor progression-free survival in patients with rectal cancer.

The authors of the second article published in Rectal Cancer [21] identified 15 metabolite biomarkers to predict the pathological response to nCRT in patients with locally advanced RC. These biomarkers were selected from a large dataset of serum samples from patients who underwent nCRT followed by rectal resection with total mesorectal excision. The biomarkers were then used to develop a logistic regression model that had an AUC of 0.76 for predicting the nCRT response. The authors also mapped the 15 metabolite biomarkers to the KEGG database and found that they were involved in a number of different metabolic pathways, including histidine, glycine, serine, and threonine metabolism, and even more in glycerophospholipid metabolism.

In 2018, there were 15 metabolite biomarkers identified to be able to facilitate the prediction of tumor response to nCRT in locally advanced RC [21]. The article published in *Front. Oncol.* [20] highlighted that the mechanisms regulating tryptophan (TRP) catabolism may be different between responsive and nonresponsive locally advanced RC patients. The enhanced predictive capability of the combined metabolite markers, surpassing the diagnostic performance of the commonly employed protein markers CEA and CA 19-9 in clinical settings for RC diagnosis was demonstrated in the article published in *J. Clin. Lab. Anal.* [24]. Specifically, the arginine and proline metabolism pathway, and the starch and sucrose metabolism pathway were found to be involved in the metabolic changes associated with cancer. The combination of d-glucose and d-mannose had an AUC of 0.805 for CC diagnosis. The combination of 2-aminobutanoic acid, 3-hydroxypyridine, d-glucose, d-mannose, isoleucine, l-tryptophan, urea, and uric acid had an AUC of 0.889 for RC diagnosis.

The article published in Cancers [25] provides insight into the molecular components of exosomes associated with the response of RC patients to nCRT. Exosomes contain a higher number of differentially expressed proteins than plasma, which can be objectively used to discriminate between different responses to nCRT. The differentially expressed proteins that are most discriminatory are involved in the immune response, vesicle-mediated transport, complement activation/protein activation, leukocyte-mediated immunity/neutrophil degranulation, and cholesterol metabolism. Other metabolites with discriminatory qualities are involved in energy metabolism (glycolysis, gluconeogenesis, trehalose degradation) and vitamin K metabolism. The pathways enriched in differentially expressed proteins are the complement/coagulation cascades, aminoacyl-tRNA biosynthesis, and cholesterol metabolism.

Across studies, the findings prove that the following biomarkers may be useful for predicting the prognosis of RC patients: glycine, creatine, myo-inositol, PE (34:2), PS (34:5), and sarcosine. These biomarkers could potentially be used to identify patients who are at high risk of poor prognosis and to develop targeted therapies.

## 4. Discussions

Molecular biomarkers, whether derived from tissue or blood samples, possess the capacity to accurately anticipate the response to neoadjuvant chemoradiation at an early stage, displaying adequate levels of sensitivity and specificity. However, as of now, none of these biomarkers have been implemented in clinical practice. By integrating diverse biomarkers, including imaging and clinicopathological characteristics, establishing associations with tumor biology, and conducting comprehensive validation using samples that accurately represent the heterogeneity of the disease, this will enable the development of a highly sensitive and cost-effective molecular biomarker panel. This will further support endeavors to offer personalized care for individuals afflicted by rectal malignancy [26].

The discovery of metabolites is crucial for comprehending the potential biological changes linked to the morbidity of rectal cancer and for integrating this approach into clinical practice [26]. Wang et al. [22] conducted a study on the metabolic profiling of human rectal cancer tissue using a H nuclear magnetic resonance (H NMR)-based metabolomics assay, a very sensitive and non-destructive technique for identifying biomarkers in biological systems. This study revealed the presence of several metabolites in human rectal cancer tissues and detected three alterations in these metabolites in conjunction with the progression of rectal cancer. These metabolites include lactate, threonine, acetate, glutathione, uracil, succinate, serine, format, lysine, tyrosine, myo-inositol, taurine, phospho-creatine, creatine, betaine, and dimethylglycine. Rectal cancer tissues exhibited a decrease in glucose levels, whereas lactate and serine levels were continuously increased, as anticipated due to the Warburg effect [27]. Cancer cells exhibit a preference for utilizing glucose through glycolysis to produce ATP, rather than relying on oxidative phosphorylation, even when there is an abundance of oxygen. Therefore, cancer cells increase their glucose intake to fulfil the energy demands necessary for their rapid multiplication [27,28]. In rectal cancer tissues, there is an accumulation of lactate, which is the result of glycolysis, along with a reduction in glucose. Lactate induces a persistent acidic extracellular pH in the tumor, hence promoting tumor cell invasion in vitro settings and metastasis in vivo. Furthermore, rectal cancer tissues exhibited elevated levels of serine, in addition to the upregulation of glycolysis. In a recent study [28], it was discovered that human cancer cells efficiently utilize external serine. By depriving them of serine, they activate the serine synthesis pathway, leading to an enhanced flow into the tricarboxylic acid cycle. These results provide compelling evidence that changes in serine metabolism contribute to the disruption of rectal cancer in humans [22,28,29].

Rectal cancer tissues exhibited down-regulation of creatine, dimethylglycine, and betaine. These metabolites are all engaged in choline metabolism pathway. Choline and its derivatives have a vital function in the metabolic process of phospholipids in cell membranes and are acknowledged as markers of cell proliferation. Although methylamines, which are produced during the breakdown of choline, are typically regarded as harmless, they have the capacity to induce liver cancer in rats, and a comparable mechanism might be present in humans. Therefore, the existence of methylamines may indicate a disruption in hepatic equilibrium while rectal cancer advances. The creatine/creatine-kinase (CK)/phosphocreatine system is critical for cellular energy buffering and transport, particularly in cells with high and disturbed energy metabolism [22,30].

Lactate can exert an immunosuppressive function within the tumor microenvironment, facilitating tumor growth by attracting and stimulating the function of immunosuppressive cells and molecules. Elevated lactate levels play a crucial role in the spread of tumor cells, vessels neoformation, and the development of resistance to treatment. The extensive research on tumor metabolism has revealed the significant clinical importance of lactate in colorectal cancer [31,32,33]. To investigate the attributes of lactate metabolism in colon adenocarcinoma, a total of 245 genes related to lactate from 13 different metabolic pathways were combined and evaluated. The study revealed 27 lactate genes that are expressed differently between tumor samples and the surrounding tissues. Amongst them, two genes connected to lactate, namely secreted phosphoprotein 1 (SPP1) and MYC proto-oncogene (MYC), are significantly increased by more than four times in the tumor samples. This study provided evidence that the upregulation of SPP1 exacerbates the prognosis of CRC patients by facilitating tumor evasion from immune responses. In brief, the lactate score can be utilized to assess the prognosis of colorectal cancer patients, as lactate levels have an impact on tumor immune infiltration and mutation burden [31].

Wu et al. employed gas chromatography-mass spectrometry (GC-MS)-based metabolomics to investigate the metabolic profile of patients diagnosed with rectal cancer. The combination of metabolite markers, including d-glucose and d-mannose for colon cancer diagnosis and 2-aminobutanoic acid, 3-hydroxypyridine, d-glucose, d-mannose, isoleucine, l-tryptophan, urea, and uric acid for rectal cancer diagnosis, showed superior predictive performance compared to CEA and CA 19-9 [24].

Li et al. created a biomarker panel composed of three proteins: c-MYC, proliferating cell nuclear antigen (PCNA), and tissue inhibitor of metalloproteinases 1 (TIMP1). It is worth mentioning that the combination of the biomarker panel with an imaging feature, namely MRI-detected extramural vascular invasion, greatly enhanced the accuracy of predicting overall survival in rectal neoplasm, thus indicating that biomarkers can enhance radiological characteristics [34,35].

Diagnostic models for colon cancer and rectal cancer were developed using the differential metabolites detected in the metabolomics data. ROC studies demonstrated that the resulting prediction models were superior to the routinely used clinical markers CEA and CA 19-9 [18]. The diagnostic specificity of CEA and CA 19-9 is limited, since they exhibit alterations not only in patients with gastrointestinal cancer but also in individuals with other malignancies, including breast and lung cancer [8]. Metabolomics is an approach that allows for the comprehensive analysis of altered metabolites in a certain state, hence identifying metabolite markers that are highly correlated with the disease under investigation. A single marker alone may not be enough to develop a test with the desired sensitivity and specificity for CRC, since it may be readily influenced by several variables. This implies that a combination of numerous metabolite markers can more accurately represent the true condition of patients [5].

Previous metabolomics investigations of CRC have identified several metabolites that exhibited inconsistent alterations in individuals. For instance, Mirnezami et al. [29] and Chan et al. [17] discovered that glycine levels were higher in patients with CRC compared to healthy individuals. Conversely, Ma et al. [27] observed a decrease in glycine levels. In this investigation, the concentration of glycine was shown to be elevated in the serum of CC patients, whereas it was reduced in the serum of RC patients.

Moreover, our findings indicated that the modification of tryptophan in CC patients was contrary to that seen in RC patients, which contradicts the findings of earlier investigations. The prior CRC investigations failed to differentiate between individuals with CC and RC, perhaps resulting in inconsistent outcomes due to variations in the distribution of these two categories. Hence, it is essential to segregate the two categories of patients and conduct extensive sample validation in future research endeavors [36,37].

In Sara Crotti’s article [20], the plasma analysis revealed the first distinction between patients who responded to treatment (TRG 1–2) and those who did not (TRG 3–5), as shown by the variations in TRP levels. Furthermore, TRG 3–5 patients exhibited elevated activity in the kynurenine pathway, which is associated with the overexpression of TDO2. Nevertheless, incongruous findings were derived from the examination of the serotonin system. The reduction in tryptophan hydroxylase 1 (TPH1) activity, as measured in both plasma and tissues, exhibited contrasting outcomes in relation to tissue expression. This likely indicates the existence of posttranscriptional control in the amount of TPH1 protein, which subsequently impacts its function in patients with TRG 3–5. It is worth mentioning that the posttranscriptional control of TPH1 and the diurnal change of TPH1 activity in the central nervous system have been shown; although, there is no known evidence regarding their association with cancer. Overall, our findings suggest that the processes controlling the breakdown of TRP may vary between locally advanced rectal cancer (LARC) patients who respond to treatment and those who do not.

Huixun Jia’s [21] study focused on the metabolomics analyses that was conducted on blood samples from patients successfully differentiated between those who were susceptible to neoadjuvant chemoradiation and those who were resistant. This enabled the creation of a novel prediction tool for LARC using a serum metabolite test. The findings of this prospective investigation demonstrated that 15 distinct metabolites had significant differences in expression, making them suitable markers for identifying LARC patients who are responsive to neoadjuvant chemoradiotherapy. Consequently, this approach might lead to favorable functional and clinical outcomes, obviating the need for full resection. On the other hand, precisely identifying patients who are unresponsive to neoadjuvant chemoradiation would reduce the likelihood of their undergoing ineffective nCRT treatment. Alternatively, it would be more beneficial to transition these individuals towards surgical intervention. The strategy suggested in the current study has significance in clinical practice as cancer research progresses towards personalized therapy.

Urszula Strybel [25] studied in her research that some molecules were linked to many key pathways involved in responding to the medication, such as the immune system, complement activation cascade, platelet activities, lipid metabolism, and glucose metabolism. Furthermore, the proteome component of exosomes exhibited the greatest number of molecules with substantially varying amounts across excellent and bad responders. This indicates that this specific fraction of blood has a strong ability to differentiate between patients who reacted differently to neo-RT. Therefore, the proteome components found in serum-derived exosomes have the potential to serve as biomarkers for predicting the response to neoadjuvant therapy in patients with rectal cancer. In addition, the incorporation of metabolomic and proteomic data provides a new and valuable understanding of the function of exosomes in relation to cancer therapy.

Glycine is a prevalent amino acid and plays a crucial role in various metabolic and pathophysiological processes that are vital for the growth and survival of rapidly dividing cells, including cancer cells. Previous studies have demonstrated a substantial correlation between higher proliferation rates and both glycine consumption and expression of the mitochondrial glycine synthesis pathway. Moreover, by antagonizing the process of glycine absorption and its mitochondrial production, the cells that were undergoing rapid proliferation were specifically inhibited. Elevated levels of the mitochondrial glycine metabolism enzymes in breast cancer are linked to increased mortality. The findings suggest that tumors’ heightened need of glycine could serve as a metabolic weakness that can be specifically targeted to hinder the rapid growth of cancer cells [23,37,38]. Kathrine Roe Redalen conducted an analysis of metabolic profiles in tumor samples from patients with locally advanced rectal cancer. These patients have undergone combined-modality neoadjuvant treatment followed by radical surgery. Redalen’s findings revealed the first clinical evidence that a high glycine content in tumors is associated with a negative prognosis for progression-free survival (PFS). Attempts to locate a sufficiently big validation group, comprising tumor tissue samples at the outset and long-term outcome data to be collected in the future, have not been successful. However, these initial discoveries have generated new and captivating ideas, which act as a motivating factor for further investigations on glycine as an early predictor of metastatic progression and a focus for treatment [23].

The study of cancer biomarkers has made significant strides in the recent years, offering valuable insights into cancer diagnosis, prognosis, and treatment response. Cancer biomarkers come in various forms, including genetic mutations, protein expressions, circulating tumor cells, and metabolites.

Despite significant progress, challenges remain in the validation and standardization of biomarkers, marking the limitations of the present studies. Biomarker heterogeneity, variations in assay techniques, and the need for large-scale clinical validation are ongoing issues. Rigorous testing through well-designed clinical trials is essential for validating the utility of biomarkers in routine clinical practice.

## 5. Conclusions

In conclusion, the study of cancer biomarkers continues to shape the landscape of cancer care, offering new avenues for early detection, personalized treatment, and improved patient outcomes. As research progresses, the integration of validated biomarkers into routine clinical practice holds the potential to revolutionize cancer management. However, ongoing efforts are needed to address challenges and ensure the translation of promising biomarkers from research laboratories to bedsides.

## Figures and Tables

**Figure 1 ijms-25-02381-f001:**
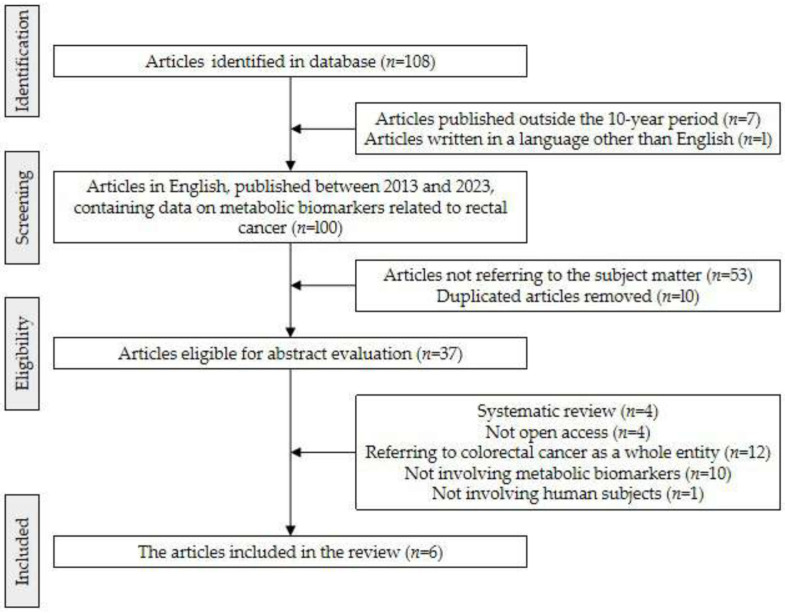
Prisma flow diagram for the selected studies included in the systematic review (records identified from PubMed, Embase, and World of Science database).

**Table 1 ijms-25-02381-t001:** Available patient demographics.

PMID Type of Study	Categories	No of Subjects		Age, Mean/Median (Range)	TRG *		Clinical T Stage				Clinical N Stage		
		M *	F *		0–2	3–5	T1	T2	T3	T4	N0	N1	N2
32281150 [24] Case-control	Control	31	14	66.49 (49–84)									
	RC	16	7	68.48 (54–80)									
	CC	15	7	64.41 (49–84)									
33178611 [20] Cohort		52	30	66 * (31–79)	37	45							
30041962 [21] Cohort	nCRT sensitive	41			56				48	8	1	24	31
	nCRT resistant	28			43	6			42	7	2	19	28
35205741 [25] Cohort	TRG * 0–1	12	5	64.9				5	11	1			
	TRG * 2–3	13	10	66.5				8	14	1			
26705680 [23] Cohort		35	19	61 (31–73)	42	12		4	31	19	7	9	38
24138801 [22] Case-control	RC	69	58	55 (28–86)			10	30	86	1	73	33	21
	Control	16	27	56 (35–85)									

* M = Male, F = Female, TRG = Tumor regression grade, nCRT = neoadjuvant chemoradiation.

**Table 2 ijms-25-02381-t002:** Biomarkers described to have predictive values through the selected research papers.

PMID	24138801 [22]	26705680 [23]	30041962 [21]	33178611 [20]	32281150 [24]	35205741 [25]
2-aminobutanoic acid					x	
3-hydroxypyridine					x	
3-methylhistidine			x			
4-Imidazoleacetic acid			x			
Acetate	x					
Betaine	x					
C8G						x
CFHR3						x
Creatine	x	x				
D-glucose					x	
Dillapional			x			
Dimethylglycine	x		x			
D-mannose					x	
Formate	x					
Ganglioside GT1b (d18:0/14:0)			x			
Glutathione	x					
Glycine		x				
GPLD1						x
Isoleucine					x	
Kynurenine				x		
Lactate	x					
L-tryptophan					x	
Lysine	x					
Myo-inositol	x	x				
N-Methylethanolamine phosphate			x			
Oleanolic acid acetate			x			
PC (9:0)			x			
PC (16:0/18:1)			x			
PC (25:1(oh))			x			
PC (27:1(oh))			x			
Phosphocreatine	x					
Serine	x					
Serotonin				x		
SERPINF2						x
SM (d18:2/14:0)			x			
Succinate	x					
Taurine	x					
Threonine	x					
Tryptamine				x		
Tyrosine	x					
Uracil	x					
Urea					x	
Uric acid					x	
δ-Valerolactam			x			

C8G = complement C8 gamma chain; CFHR3 = complement factor H related 3; GPLD1 = glycosylphosphatidylinositol specific phospholipase D1; PC = phosphatidylcholine; SERPINF2 = serpin family F member 2; SM = sphingomyelin, x = evaluated in the specific study.

## Data Availability

Data is contained within the article and available on request from the corresponding author.

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
