# Peer review of "Metabolic Signatures: Pioneering the Frontier of Rectal Cancer Diagnosis and Response to Neoadjuvant Treatment with Biomarkers—A Systematic Review"

_ijms, 2024, doi:10.3390/ijms25042381_

Round 1

Reviewer 1 Report

Comments and Suggestions for Authors

The research paper under review references a limited number of studies, which are notably diverse, despite their shared focus on the investigation of metabolites in rectal cancer. The results section lacks commentary on both the cohorts selected and the markers established. Furthermore, there is an absence of analysis concerning the common and distinct metabolites identified across the individual studies.

This part contains elements of discussion and commenting on foreign research. 

The research lacks depth and sounds too general in places, especially in the case where the authors tried to make general conclusions and indicate key points regarding the study of cancer biomarkers.

Overall, the paper would benefit from a more focused and detailed approach to enhance its contribution to the field of cancer biomarker research.

Author Response

Thank you for the kind remarks, we greatly appreciate your comments and all the requirements were thoroughly addressed:

The number of studies included in the present review is limited due to the scarce amount of articles regarding this theme, available in the literature, even less of a satisfactory quality of research. Therefore, we selected only the most comprising and sound ones in relationship with the scope of our study.

In the Results section, which we improved for better outcome, paragraph 1 to 6 presents now in extenso the cohort and metabolites selected and for emphasizing visual impact Table 1 presents the cohort for each study and in Table 2 the metabolites evaluated/ study.

The elements containing general information at the end of the Results section was excluded, and the important aspects from there added in the Discussion section.

We have improved our Results section based on the remarks obtained and pointed their value in a more focused way.

Reviewer 2 Report

Comments and Suggestions for Authors

In this manuscript, the authors provide a systemic review of published papers aiming to describe possible metabolic biomarkers or a sequence of metabolomic indicators with diagnostic value for predicting the efficacy of neoadjuvant therapy. I think this is an interesting topic and it is of great importance for the patients. The paper is generally well written, and the introduction and the discussion sections provide sufficient information. My main concern is the results section which is poorly written and provides very little information. My recommendations are numbered below.

1. Please make sure that all abbreviations (like CC, RR, etc.) are clarified when they first appear in the beginning of the manuscript (preferably introduction). Also clarify abbreviations used in tables as there are no table legends with descriptions.

2. Could the authors improve the resolution of figure 1 please.

3. Table 2 has misaligned markings. Please align “x” to where it belongs.

4. In the results section, the authors place two tables and talk very little about those results. Could the authors talk more about the results, also highlighting the most important ones.

5. In the results section, the authors provide a bulleted list of general conclusions which don’t seem to emerge from the presented results. Also, passing from narrative to a bulleted list is a bit unusual. Could the authors fix the format or explain where those results come from (if they are results) or place it under the correct section.

Comments on the Quality of English Language

There are minor errors throughout the manuscript that need to be corrected.

Author Response

Thank you for the kind remarks, we greatly appreciate your comments and all the requirements were thoroughly addressed:

We have revised the Results section and provided a more focused set of information, discarding the too general content.

  1. We have explained all abbreviations, both in text, where they first appeared (e.g. EV - extracellular vesicles and others, which lacked explanations) and in Table legends.
  2. We have improved Figure1's resolution.
  3. We have realigned the "x" where they belonged. It appeared due to the difference in format between multiple computers and programs. 
  4. We have further developed the contents of the two tables in the paragraphs between them from the Results section and emphasised the important aspects. 
  5. The elements containing general information at the end of the Results section were excluded, and the important aspects from there added in the Discussion section.

We have improved our Results section based on the remarks obtained and pointed their value in a more focused way.

The entire manuscript was thoroughly checked for language grammar and spelling mistakes, and corrected.

Round 2

Reviewer 1 Report

Comments and Suggestions for Authors

The authors have responded to my comments and recommendations. They have made significant corrections to the text, but the section titled “Results” is more of a discussion, and the authors have not yet presented their analysis of the selected publications. The manuscript should be restructured to be titled as a “Systematic Review”.

Author Response

Thank you for your constructive remarks.

We have reassembled and reorganised the Results section in order to confer a clearer and more logical approach to each subsegment of the articles analysis: number of patients, samples used and their way of preservation, methods applied for the measurement of certain biomarkers, the markers identified and their statistical power etc.

Reviewer 2 Report

Comments and Suggestions for Authors

I thank the authors for all the corrections and improvements of their manuscript. The manuscript is now ready to be published.

Author Response

Thank you for your review.